# Tailoring the Hydroxyl Density of Glass Surface for Anionic Ring-Opening Polymerization of Polyamide 6 to Manufacture Thermoplastic Composites

**DOI:** 10.3390/polym14173663

**Published:** 2022-09-03

**Authors:** Achraf Belkhiri, Nick Virgilio, Valérie Nassiet, Hélène Welemane, France Chabert, Olivier De Almeida

**Affiliations:** 1Laboratoire Génie de Production (LGP), Institut National Polytechnique de Toulouse (INP)-Ecole Nationale d’Ingénieurs de Tarbes (ENIT), Université de Toulouse, 65000 Tarbes, France; 2Institut Clément Ader (ICA), Université de Toulouse, CNRS UMR 5312, IMT Mines Albi, UPS, INSA, ISAE-SUPAERO, Campus Jarlard, 81013 Albi, France; 3Centre de Recherche sur les Systèmes Polymères et Composites à Haute Performance (CREPEC), Department of Chemical Engineering, Polytechnique Montréal, Montréal, QC H3C 3A7, Canada

**Keywords:** composite materials, particle surface modification, reactive PA6 polymer matrix, anionic ring-opening polymerisation, particle/matrix interface, hydroxyl groups, silane

## Abstract

Reactive thermoplastics matrices offer ease of processing using well-known molding techniques (such as Resin Transfer Molding) due to their initially low viscosity. For Polyamide 6 (PA6)/glass composites, the hydroxyl groups on the glass surface slow down the anionic ring-opening polymerization (AROP) reaction, and can ultimately inhibit it. This work aims to thoroughly control the hydroxyl groups and the surface chemistry of glass particulates to facilitate in situ AROP-an aspect that has been barely explored until now. A model system composed of a PA6 matrix synthesized by AROP is reinforced with calcinated and silanized glass microparticles. We systematically quantify, by TGA and FTIR, the complete particle surface modification sequence, from the dehydration, dehydroxylation and rehydroxylation processes, to the silanization step. Finally, the impact of the particle surface chemistry on the polymerization and crystallization of the PA6/glass composites was quantified by DSC. The results confirm that a careful balance is required between the dehydroxylation process, the simultaneous rehydroxylation and silane grafting, and the residual hydroxyl groups, in order to maintain fast polymerization and crystallization kinetics and to prevent reaction inhibition. Specifically, a hydroxyl concentration above 0.2 mmol OH·g^−1^ leads to a slowdown of the PA6 polymerization reaction. This reaction can be completely inhibited when the hydroxyl concentration reaches 0.77 mmol OH·g^−1^ as in the case of fully rehydroxylated particles or pristine raw particles. Furthermore, both the rehydroxylation and silanization processes can be realized simultaneously without any negative impact on the polymerization. This can be achieved with a silanization time of 2 h under the treatment conditions of the study. In this case, the silane agent gradually replaces the regenerated hydroxyls. This work provides a roadmap for the preparation of reinforced reactive thermoplastic materials.

## 1. Introduction

In composite manufacturing, several materials can be used as polymer reinforcement, such as carbon fibers, glass fibers, graphene or carbon nanotubes [1,2,3]. Long fibers offer the best mechanical properties and include glass or carbon fibers, aramid, polyethylene and cellulose fibers. Glass fibers are also among the most commonly used reinforcements, offering a good compromise between satisfying mechanical properties and ease of manufacturing [4]. The mechanical behavior of glass fiber-reinforced polymer composites highly depends on the applied surface treatment and sizing of the fibers [5]. Sizing plays a crucial role in load transfer at the interface between the polymer matrix and fibers. However, the nature of the sizing and its chemical composition remain poorly documented in the literature, most often due to confidentiality reasons. Indeed, chemical surface treatments of solid fiber and particulate reinforcements represent a strategic aspect for several industrial applications requiring strong mechanical properties, including the aeronautics and automotive sectors [6].

A sizing formulation is typically an aqueous emulsion containing a coupling agent (<1%), a film former (4–7%), a lubricant (≈0.1%) and an electrostatic agent (≈0.1%) [7].

Fiber/matrix adhesion is mainly governed by the coupling agent [8], and is clearly enhanced when the coupling agent interacts with both the matrix and the reinforcing particles [9,10]. In the case of glass fibers, the most commonly used coupling agents are organosilanes, which form covalent bonds by reacting on one side with the silanol groups at the fiber (or particulate) surface, via condensation reactions after hydrolysis of the alkoxy groups [7]. Silanes can also contain complementary functional groups creating bonds with the polymer matrix [5]. The surface density of the grafted silane agent is a particularly important parameter since it strongly impacts the composite properties [11,12]. The grafting degree mainly depends on the glass fiber’s initial hydroxyl groups surface concentration, corresponding to the reaction sites for the organosilane’s silanol groups [7]. Consequently, controlling the initial hydroxyl surface density represents a key process for the manufacturing of optimized composite materials.

While industrial sizings are typically well suited for most commercial polymeric matrices, surface treatment must still be adapted to specific or new polymers. A typical case requiring careful and precise control over sizing is for new reactive thermoplastic matrices developed to overcome the high viscosity of commercial thermoplastics. These matrices can be processed by conventional composites molding techniques, such as Resin Transfer Molding (RTM), which are directly polymerized in the mold. For the specific case of polyamide 6 (PA6) matrices obtained by the anionic ring-opening polymerization of ϵ-caprolactam monomers, the reactive mixture also comprises a catalyst and an activator, which react in contact with the glass fiber surface [13]. In this case, controlling the hydroxyl density on the glass surface is crucial because it directly impacts the polymerization reaction. Indeed, the hydroxyl groups on the glass surface slow down the polymerization reaction, and can ultimately inhibit it, due to the labile protons that deactivate the catalyst (Figure 1). This can also lead to a significant reduction in the degree of monomer conversion [13]. For this reason, carefully controlling the grafting of an organosilane at the glass surface, before composite manufacturing, is essential in order to enhance the final mechanical properties, while simultaneously limiting the negative effect of the possible residual hydroxyl groups on the polymerization reaction [13].

Alkoxysilane coupling agents are usually grafted at the surface of glass fibers from an aqueous solution, or sometimes from an organic solvent when the silane is not soluble in water [14]. The aqueous solution allows the hydrolysis of the silane and the formation of silanol groups. Initially, the formed silanol groups interact with the hydroxyl groups on the glass surface via hydrogen bonds. Then, the condensation of these moieties generates siloxane bonds [9]. However, treatment in an aqueous solution can also lead to rehydroxylation of the glass surface, and regeneration of the hydroxyl groups from the siloxane groups [15,16]. As a result, rehydroxylation increases the surface density of hydroxyl groups, impacts the grafting degree of organosilanes, and ultimately increases the residual hydroxyl groups’ surface density. Therefore, grafting silanes on glass fibers is clearly a delicate and complex dynamic process resulting from a competition between the condensation of the silanes on the surface, which decreases the number of -OH groups, and rehydroxylation, which leads to regeneration and an increase in the surface concentration of these groups. This competition depends on several factors including the pH and silanization time [9,14].

In order to understand and quantify the influence of these parameters on the glass surface treatment, and to design the most appropriate process (temperature, time, etc.) to optimize the hydroxyl groups’ surface density, it is necessary to monitor the density of -OH groups at the surface. For this purpose, adequate calcination treatments are suitable for the nearly complete removal of hydroxyls from the surface, the evolution of the hydroxyl density depending on the calcination temperature and time. The impact of the calcination temperature on the hydroxyl groups’ surface density has been investigated by some authors [15,17,18], while the influence of the calcination time remains much less understood.

Different techniques have been used to quantify the hydroxyls surface density, such as deuterium exchange [15,18], infrared spectroscopy [19,20] or nuclear magnetic resonance spectroscopy (NMR) [21]. These techniques typically provide comparable density values. Thermogravimetric analysis (TGA) is another relevant technique to monitor the dehydration and dehydroxylation processes, and thus to determine the surface density of hydroxyl groups [22,23]. Once dehydration is completed, the loss of physically adsorbed water can be subtracted from the total loss to estimate the number of surface-bound hydroxyl groups. TGA can also be coupled with mass or IR spectroscopy for better precision in the evaluation of the hydroxyl groups’ surface density, and to distinguish the dehydration from the dehydroxylation processes [24]. Kellum et al. [22] compared the amounts of physically adsorbed water, and surface-bound -OH, using different techniques. The -OH densities calculated from the TGA results were quite comparable to values obtained from other conventional methods, confirming that TGA can provide accurate values. Mueller et al. [23] also highlighted the TGA reliability and reproducibility to determine the surface density of hydroxyl groups. Moreover, as physically adsorbed water can be separated from silanol condensation with TGA, it allows an accurate estimation of the -OH density value. Finally, it was demonstrated that the weight loss induced by physically adsorbed water is not impacted by the drying conditions during dehydration so the hydroxyl density was not affected either. These results confirm that it is possible to distinguish both the dehydration and dehydroxylation processes by TGA [23].

In order to improve the properties of reactive thermoplastic composite materials, optimizing the surface treatment of particulate or fiber reinforcements must therefore be realized in association with adequate control over the polymerization process of the matrix. While the effect of calcination temperature on the hydroxyl groups’ surface density has been extensively investigated, special attention is dedicated in this work to the effect of calcination time. In addition, understanding the effect of the silanization treatment time, and the competition between the condensation and rehydroxylation reactions during the silanization process in an aqueous solution, is needed.

This work intends to bring a better understanding regarding the surface treatment protocols for the preparation of glass composite reinforcements, by means of systematic and rigorous monitoring of the -OH surface density and concentration during the silanization steps. This methodology is implemented for the case of glass microparticles, a model system allowing to remove the effects of fiber length and orientation, in order to specifically focus on the impact of surface chemistry, by using techniques such as TGA coupled with IR spectroscopy. Finally, the impact of the silane grafting protocol is evaluated on the polymerization and crystallization kinetics of a model system composed of a PA6 matrix synthesized by anionic ring-opening polymerization, reinforced by silanized glass microparticles, by means of DSC measurements. In our application, the control of the hydroxyl groups amount allows us to preserve the polymerization and crystallization kinetics during the “in-situ” synthesis of anionic PA6-based composite. Many other applications can benefit from improved -OH groups control using the same protocol, such as composites manufacturing, glass coatings for hydrophobic surfaces, antibacterial and antifungal surfaces, waterproof windows, antifouling coatings for eyeglasses and any other application requiring silane treatment.

## 2. Materials and Methods

### 2.1. Materials

Glass microparticles were purchased from the Sovitec company (France) with an average diameter of 4 μm and a particle size distribution D50 of 3.68 μm. This particle size was chosen according to two criteria: (1) the diameter was chosen in order to have a specific surface area equivalent to glass fibers (diameter of 5 μm) distributed in a composite, at an equivalent fiber volume contents of 60%; (2) the volume fraction of microparticles should not exceed 30% in order to facilitate the mixing process during manufacturing, and to ensure the homogeneity of the mixture.

The monomer (ϵ-caprolactam (CL) “AP-Nylon”), the catalyst (caprolactam magnesium bromide MgBrCL-“Nyrim C1”, 1.4 mol·kg^−1^ in caprolactam) and the activator (bifunctional hexamethylene-1,6-dicarbamoylcaprolactam (HDCL) “Bruggolen C20P”, 2.0 mol·kg^−1^ in caprolactam) used for the anionic ring opening polymerization of PA6 (see reaction mechanism in Appendix A) were all supplied by the Brüggemann Chemical company (Germany). Since storage and processing have to be conducted in a moisture-free environment due to the sensitivity of the reaction to water, the products were dried overnight at 30 °C under a vacuum before each synthesis. Then, all handling was realized in an inert atmosphere in a glove box. The mixture was prepared first by adding the monomers in a beaker at 70 °C with stirring on a magnetic hot plate. After total melting of the monomers, the catalyst was added, followed by the activator after total melting of the catalyst, under moderate stirring. The formulation of MgBrCL/HDCL used was 0.79/1.10 mol% of CL. Then, the mixture was quenched in liquid nitrogen to prevent initiation of the reaction, before storing in a hermetic container at −18 °C. Finally, 3-(2-aminoethylamino)propyltrimethoxysilane (AEAPTMS, 96%) was supplied by Fisher Scientific.

### 2.2. Particle Surface Modification

#### 2.2.1. Control of the Surface Density of Hydroxyl Groups

The particles were first calcinated at 450 °C for 2 h to remove impurities from the surface. Then, they were rehydroxylated for 8 h in a 10% (*v*/*v*) aqueous hydrochloric acid solution, under stirring at room temperature, to saturate the surface with hydroxyl groups. These particles are designated as *fully rehydroxylated particles*. After complete rehydroxylation, the particles were washed several times with distilled water until the pH stabilized at a value of 7.0, before drying at 115 °C for 2 h. Samples of these fully rehydroxylated particles were then calcinated at 450 °C for up to 24 h in a TGA instrument, in order to follow the evolution of the hydroxyl groups surface density as a function of calcination time. This calcination temperature was chosen following the work of Young [17], who demonstrated that it maximizes the formation of reversible dehydroxylated sites on glass surfaces. These sites allow control over the rehydroxylation and silane grafting processes, which cannot be realized with irreversible dehydroxylated sites obtained at higher temperatures.

#### 2.2.2. Optimization of the Treatment Time

Once a suitable calcination time was determined, the rehydroxylation process was evaluated by monitoring the regeneration of hydroxyl groups at the surface of the calcinated particles when treated in the silanization aqueous solution (composed of distilled water adjusted at a pH of 4–5 with acetic acid, but without the silane agent). Glass particles were treated in this solution for 2, 3 or 6 h, respectively, at room temperature. Silanization in acidic conditions is relevant to promoting the formation of silanol groups and slowing down the self-condensation reaction between the resulting hydrolyzed silanol groups [25]. Then, the particles were rinsed twice with ethanol, twice with water, and dried for 2 h at 115 °C.

#### 2.2.3. Surface Modification by Silanization

First, an aqueous solution of AEAPTMS (5% *v*/*v*) was prepared. The pH was adjusted to 4–5 with acetic acid. The mixture was stirred for about 45 min. After hydrolysis of the silane, 50 g of calcinated glass particles were added to 100 mL of solution with stirring for 1 h (or 3 h). The mixture was then heated for an additional 1 h (or 3 h) at 100 °C to condense the silanol groups on the surface and to remove the traces of methanol from the hydrolysis of the methoxysilane. The particles were finally rinsed twice with ethanol and twice with water to remove the unreacted silanes, then dried for 2 h at 115 °C (see mechanism in Appendix A).

### 2.3. Particle Surface Characterization

#### 2.3.1. Specific Surface Area Measurement

The specific surface area was measured with a 3Flex BET instrument from Micromeritics. This apparatus measures the volume of adsorbed gas (nitrogen) on the surface of the samples at the temperature of liquid nitrogen (77 K) up to a relative pressure of 1 in order to plot a complete adsorption isotherm. A preliminary degassing at 50 °C for 11 h was performed to clean the surface of the samples from any molecules that could obstruct access to the pores. The specific surface area was then calculated by using the BET model in a suitable pressure range.

#### 2.3.2. Thermogravimetric Analysis (TGA)

The TGA experiments were realized with a TGA1 STARe System-METTLER TOLEDO. First, the weight loss due to physically adsorbed water was determined by heating the glass particles from 25 °C to 150 °C at 10 °C/min in an air atmosphere. The temperature was then held at 150 °C for 2 h. This temperature was chosen as a mean value based on the literature [15,22,23]. The resulting weight loss was subtracted from the total weight loss obtained after each test, according to previous works [22,23].

Next, the glass particles were heated from 25 °C to 450 °C at 10 °C/min in an air atmosphere. The temperature was maintained at this value until the sample mass stabilized. As explained previously, this maximum temperature was chosen in order to obtain a maximum number of dehydroxylated sites that could be reversibly rehydroxylated. Above this temperature, the number of hydroxyl sites that can be regenerated decreases, which lowers the total number of available hydroxyl sites after rehydroxylation, which afterwards affects control over the silane grafting process. More specifically, this temperature prevents the removal of all silanols from the surface and thus allows silane grafting. Indeed, above this temperature, the condensation of silanols becomes more important and irreversible, leading to the irreversible loss of most (or all) hydroxyl groups, subsequently inhibiting silane grafting [17,18].

After silanization, the treated particles were characterized by TGA, in order to quantify the amount of grafted silane, and to calculate the grafting concentration and density, following the same calcination protocol as described above. Air was used as an analysis purge gas in order to accelerate and ensure complete silane degradation before reaching the calcination temperature employed in this work (450 °C), allowing us to subsequently quantify the total amount of grafted silane [26].

#### 2.3.3. Ft-Ir Spectroscopic Measurements

Transmission Fourier Transform Infrared (FTIR) spectra were acquired with a Bruker Vertex 70 spectrometer. The particles were blended with high purity infrared grade KBr powder at 1.5–2 wt% and pressed into pellets for measurements. The spectra were recorded between 400 cm^−1^ and 4000 cm^−1^ with a resolution of 2 cm^−1^. Before measurement, a background was obtained with a pure KBr pellet.

### 2.4. Differential Scanning Calorimetry (DSC) Characterization of PA6 Polymerization and Crystallization

#### 2.4.1. Sample Preparation

DSC pans were systematically prepared with mixtures of reactants and glass particles previously dried overnight at 30 °C under vacuum. The aluminum pans were filled and hermetically sealed under an inert atmosphere, to prevent monomer evaporation and moisture absorption. In this way, moisture uptake was prevented before polyamide synthesis, and glass surface rehydroxylation due to moisture was avoided. As a reference, neat resin samples were prepared by adding 4–10 mg of the reactive mixture in the DSC pans. In the case of glass particles/PA6 composites, a small quantity of particles was poured into the pan first. Then, the reactive mixture was added to reach a content of 30 ±3% vol. in glass particles, for a total weight of 10–12 mg. This constant composition ensured performing PA6 synthesis under similar conditions.

#### 2.4.2. DSC

Reaction kinetics were investigated under isothermal conditions. All samples were first heated at 300 °C/min from 25 °C to $T_{iso}$ = 180 °C. Then, they were maintained at $T_{iso}$ for 50 min. This temperature was chosen based on the work of Vicard et al. [27], who showed that this synthesis temperature allows us to decouple the polymerization and crystallization processes and, therefore, to identify their respective kinetics. Finally, the samples were cooled to 0 °C at −10 °C/min. After isothermal synthesis and cooling, each sample was heated from 0 °C to 270 °C at 10 °C/min to obtain the melting temperature and enthalpy of the crystalline phase (1st heating. Then, in order to analyze the crystallization behavior of the polymerized matrix, the DSC samples were cooled again at −10 °C/min down to 0 °C, and finally heated at 10 °C/min to 270 °C (2nd heating).

### 2.5. Composite Mechanical Properties

In order to assess the benefits of the proposed silanization protocol, composite specimens were fabricated and the tensile properties were characterized. Tensile test specimens were manufactured by directly pouring the reactive mixture containing the treated particles into a mold in accordance with the ASTM D638 type 1 tensile geometry and cured at 180 °C. The tensile tests were performed using an Instron tensile machine equipped with a 50 kN load cell with a cross-head speed of 5 mm/min. The standard method recommends a thickness below 7 mm. In our case, an average thickness of 3.2 ± 0.5 mm was measured for all specimens.

All specimens were dried for at least 4 h at 70 °C before trials. A set of four composite samples of each surface chemistry was tested at 23 °C.

## 3. Results

### 3.1. Characterizing and Controlling the Hydroxyl Groups Surface Density on Glass Particles

#### 3.1.1. Effect of the Initial Calcination Time on the -OH Surface Density

In order to calculate the hydroxyl surface density on glass particles by TGA, it is necessary to determine the mass loss specifically related to the dehydroxylation process, by excluding the mass loss related to the physically adsorbed water. The amount of adsorbed water is obtained by heating the particles for 2 h at 150 °C. For the fully rehydroxylated particles, the associated mass loss during this heating step is shown in Figure 2. The TGA results show that the percentage of physically adsorbed water is 0.34%. During this step, dehydroxylation does not occur because the associated onset temperature (200 °C) is not yet reached [15,18]. Thus, this percentage was subtracted from the total mass loss after the dehydroxylation heating cycle, which then allows the calculation of the hydroxyl surface density.

Figure 3a shows the TGA total mass loss for the fully rehydroxylated particles as a function of time, at 450 °C. The mass loss stabilizes after approximately 24 h, which corresponds to maximum dehydroxylation. Subtracting the mass loss due to the physically adsorbed water from the total mass loss gives the amount related only to dehydroxylation, and thus leading to the hydroxyl groups surface density $d_{OH}$ (in OH·nm^−2^) given by Equation (Equation 1) :(1)$$d_{OH}=\frac{2*(\frac{wt\%}{100})*Na}{M*S}$$
in which (wt%) is TGA weight loss percentage, Na is Avogadro’s constant, *M* is the molecular weight of water (g·mol^−1^) and *S* is the specific surface area of glass particles, obtained by BET.

Equation (Equation 1) yields a hydroxyl surface density value of 460 OH·nm^−2^ for the fully rehydroxylated particles. Yet, it has been reported in the literature that the hydroxyl density on a glass surface cannot exceed 6 OH·nm^−2^[15]]. However, several authors have shown that it is quite possible to find much higher values, mostly because of the specific surface measurement technique [28,29,30]. Indeed, the specific surface considered in the calculation was obtained by the BET technique. This method is based on the adsorption of nitrogen gas and is often criticized since a large amount of microporosities are not accessible to the nitrogen molecules at 77 K, and as a result are not considered in the measurement [31,32,33,34]. The specific surface area of glass particles is thus underestimated, leading to very high values of hydroxyl densities (Appendix A).

Although the calculated hydroxyl density value for the fully rehydroxylated particles is quite high at 460 OH·nm^−2^, the number of hydroxyl moles per unit of particles mass $C_{OH}$ for the fully rehydroxylated particles is 0.77 mmol OH·g^−1^ according to Equation (Equation 2), which is consistent with the literature for similar size particles [18,28]. In order to avoid the uncertainty due to the BET specific surface measurement, the hydroxyls mass concentration value of the fully rehydroxylated particles (0.77 mmol OH·g^−1^) will be considered, in this work, as a reference value that will be used for comparison purposes. From this reference value, which is proportional to the hydroxyl groups’ density, it will be indeed possible to follow the evolution of hydroxyls on the surface after each treatment.
(2)$$C_{OH}=\frac{2*(\frac{wt\%}{100})}{M}$$
in which (wt%) is TGA weight loss percentage and *M* is the molecular weight of water (g·mol^−1^).

Figure 3b illustrates the evolution of the mass loss and, consequently, the hydroxyl groups’ mass concentration remaining at the surface, as a function of calcination time at 450 °C. The results confirm the gradual decrease in hydroxyl groups at the surface, with a clear trend modeled by Equation (Equation 3) :(3)$$C_{OH}=-0.474log\left(t\right)+0.64$$
in which $C_{OH}$ (OH·nm^−2^) is the hydroxyl groups’ mass concentration, and *t* is the calcination time. Considering the good fit between the experimental results and the linear regression model (Figure 3b), it is thus possible to estimate the required calcination time for a targeted hydroxyl surface concentration.

The FTIR spectra of the fully rehydroxylated particles as well as those of particles fully rehydroxylated and subsequently calcinated from 2 h to 24 h are illustrated in Figure 4a. The results are focused on the main peak in-between 3430 cm^−1^ and 3500 cm^−1^ attributed to the hydroxyl groups since no other significant change was noticeable over the entire FTIR spectra (Appendix A). The intensity of the peak decreases progressively by increasing the calcination time, until it almost disappears after 24 h of calcination. Such a decrease confirms the progressive surface dehydroxylation, and is consistent with the gradual mass loss observed by TGA. The transmittance peak *T* (%) as a function of calcination time *t* (h) can be fitted quite well with a linear regression (Equation (Equation 4)):(4)T=12.55log(t)+80.26

By combining Equations (Equation 3) and (Equation 4), it is then possible to establish an interrelationship between the hydroxyl surface concentration and the transmittance (Equation (Equation 5)):(5)$$C_{OH}=-0.04T+3.67$$

The fully rehydroxylated particles show a peak with a broad band around 3432 cm^−1^. This peak gradually shifts to a higher wavenumber and its width decreases with increasing calcination time, up to 3500 cm^−1^ for the particles calcinated for 24 h. Indeed, the wavenumber strongly depends on the interactions between the hydroxyl groups and their environment. When hydroxyl groups form hydrogen bonds, the associated wavenumber decreases [29]. For the fully rehydroxylated particles, the hydroxyls are sufficiently close to each other to form hydrogen bonds. When the hydroxyls are removed by increasing the calcination time, the -OH groups remaining at the surface become progressively isolated, which increases the wavenumber.

#### 3.1.2. Impact of Dehydroxylation on PA6 Polymerization

Composites were synthesized and analyzed by DSC with the fully rehydroxylated particles, and with particles calcinated from 2 h to 24 h, in order to determine the effect of calcination time on the polymerization and crystallization kinetics of PA6. The results are compared with the neat resin.

The DSC thermograms of the composites in Figure 5 display two peaks: the first one is related to monomer polymerization, and the second one is related to the crystallization of PA6 [27]. Compared to the neat resin, the fully rehydroxylated particles completely inhibit the reaction. The same result was obtained with the raw pristine particles (as received and without further treatment), which confirms the incompatibility of the raw glass surface with the synthesis of anionic PA6 due to the significant presence of -OH groups.

In contrast, the polymerization and crystallization levels significantly increase by increasing the calcination time up to 8 h, which then stabilizes for longer calcination times. This improvement is due to surface dehydroxylation and to the partial removal of hydroxyl groups during calcination, which limits the deactivation of the catalyst by labile protons. The stabilization of the polymerization/crystallization kinetics after 8 h of calcination time indicates that the remaining -OH surface concentration has no significant influence on the polymerization/crystallization phenomena. The polymerization/crystallization kinetics do not change significantly when the calcination time went beyond that duration and the difference in kinetics compared to the pure resin is due to the hydroxyls that remain on the surface.

Calcinating for at least 8 h yields a hydroxyl surface concentration of about 0.20 mmol OH·g^−1^ (from the results in Figure 3b), with a minimum impact on the polymerization and crystallization phenomena. From this point, this threshold surface concentration will be identified as $C_{limit}$.

After calcinating for 10 h, the residual hydroxyl surface concentration is about 0.17 mmol OH·g^−1^, according to Equation (Equation 2). This last condition is an adequate compromise that provides some level of adjustment, considering the following silane grafting process in an aqueous solution that leads to surface rehydroxylation. Ultimately, the aim is to optimize the treatment time without exceeding the number of surface -OH groups $C_{limit}$, in order to avoid a negative impact on the polymerization/crystallization kinetics.

### 3.2. Controlling the -OH Quantity Regenerated during the Silanization Treatment

When silanization is performed in an aqueous solution, the rehydroxylation and silanization processes can occur simultaneously, which could again lead to excessive hydroxyl regeneration-especially if the silane concentration has significantly decreased due to its reaction at the particle’s surface. These regenerated and unreacted -OH could then remain at the surface and slow down or inhibit the polymerization of the PA6 matrix. Therefore, controlling the evolution of the hydroxyl groups’ mass concentration during the silanization process as a function of time is also required.

Table 1 shows the evolution of the hydroxyl groups regeneration at the particle’s surface, in the silanization solution (aqueous solution at pH 4–5 adjusted with acetic acid, without the silane at this point), as a function of time (the adsorbed water has been subtracted from the results, see the procedure in Section 3.1.1). As expected, prolonging the treatment gradually increases the surface concentration of hydroxyl groups.

The FTIR results in Figure 6b show an increase in the intensity of the peak relative to -OH groups with increasing treatment time, which confirms the regeneration of hydroxyl groups in the silanization solution. The peak intensity after 2 h of treatment is lower compared to 3 h and 6 h, but slightly higher compared to the calcinated particles, indicating moderate hydroxyl regeneration. Apart from the moderate -OH regeneration visible in the range from 3200 to 3700 cm^−1^, no other change could be noticed over the infrared range (Appendix A).

In order to assess the effect of the treatment time in the silanization solution on the polymerization/crystallization kinetics, composites were synthesized with particles first calcinated for 10 h, and subsequently rehydroxylated in the solution for 2 h, 3 h and 6 h. Figure 6b shows the DSC thermograms of the composites during synthesis. The polymerization/crystallization kinetics for the composite synthesized with calcinated particles is slightly shifted compared to the neat resin, as explained in the previous section. The polymerization/crystallization kinetics are significantly slowed down with increasing treatment time. The 2 h treatment yields the fastest kinetics among all of the treated particles conditions.

The differences observed between the composite prepared with strictly calcinated particles, and calcinated particles subsequently rehydroxylated for 2 h, is due to the regenerated hydroxyl surface concentration, which exceeds the $C_{limit}$ threshold (see Figure 5 and previous section). However, the resulting concentration does not completely inhibit the polymerization reaction. It is expected that this slight excess could be compensated by the grafting of the silane coupling agent, which is examined in the next section.

In addition, since silane grafting at a surface is limited by steric hindrance [9], the grafting density typically exhibits a maximum value [9,35]. As a result, over-prolonging the grafting time after the maximum silane surface density is reached only results in an increased number of inaccessible -OH. These inaccessible hydroxyls remain on the surface and can slow down, and ultimately inhibit, the polymerization reaction. In the present case, treating the particles in the silanization bath (without the silane) for 2 h results in moderate rehydroxylation ($C_{OH}$ = 0.21 mmol OH·g^−1^ ≈ $C_{limit}$, see previous section). Such a duration was thus chosen for the following silane grafting experiments.

Finally, we have compared the rehydroxylation process in the silanization aqueous solution, to rehydroxylation in a 10% HCl solution (typically used prior to silanization [16,36]). The aqueous silanization solution offers much more control over rehydroxylation, as compared to 10% HCl, as it avoids excessive -OH regeneration for comparable treatment times (Appendix A). Simultaneous rehydroxylation and silanization also eliminate an additional step in the surface treatment process (no acidic rehydroxylation step), while avoiding a negative impact on the polymerization/crystallization processes.

### 3.3. Polymerization and Crystallization Kinetics after Silane Grafting

#### 3.3.1. Silane Surface Modification and Its Influence on Polymerization and Crystallization

Figure 7a shows the FTIR spectra of freshly calcinated particles (450 °C, during 10 h), compared to calcinated particles subsequently silanized for 2 h. The peak related to the hydroxyls almost disappears after silanization, which demonstrates that almost all hydroxyls have reacted during the grafting process. As a result, silane grafting allows for almost complete elimination of both the residual hydroxyls remaining after 10 h of calcination, and the regenerated hydroxyls during the silanization treatment.

For the silanized particles, the mass loss measured by TGA includes both the loss of -OH groups, and the loss of the grafted silane. It is thus possible to deduce the mass loss related only to the -OH groups from Equation (Equation 5), which provides the hydroxyl groups’ surface concentration from the transmittance peak. Accordingly, Table 2 shows the total mass loss, hydroxyl mass loss, and silane mass loss, for particles silanized for 2 h, compared to the calcinated particles. The mass loss related to the -OH groups decreases considerably after silanization. These results confirm the successful grafting of the silane agent on the surface of the particles.

The mass concentration of the grafted silane ($C_{silane}$) can be calculated from the corresponding mass loss in Table 2, with Equation (Equation 6):(6)$$C_{silane}=\frac{\Delta m}{M}$$
where Δm is the mass loss associated with the grafted silane, during calcination (in Table 2) and *M* is the silane molecular weight. The calculation gives a result of about 0.05 mmol·g^−1^ of silane, which corresponds to 30 grafted silane chains per nm^2^ according to the Equation (Equation 7). This silane density value is high but remains in the order of magnitude of the results reported by some authors [37,38,39,40]. The high value could be due to an underestimation of the particulates specific surface due to the BET technique limitations, as pointed out in Section 3.1.1, and/or to the formation of silane multilayers at the particulate surface [40]. This high level of grafting is also supported by the FTIR data of Figure 7a.
(7)$$d_{silane}=\frac{C_{silane}*Na}{S}$$
where $C_{silane}$ is the mass concentration of the grafted silane, Na is Avogadro’s number and *S* the particle’s specific surface.

Next, the effect of silane grafting on the polymerization/crystallization processes was evaluated by DSC in Figure 7b. The composites were synthesized with particles silanized for 2 h. The results are compared to the neat resin, the freshly calcinated particles (10 h), and particles rehydroxylated for 2 h in the silanization solution (but without the silane agent). The thermograms confirm that the addition of the silane coupling agent accelerates the reaction, compared to the rehydroxylated particles without the silane agent. As expected, the excess hydroxyl surface concentration (compared to $C_{limit}$) was fully compensated by the addition of the silane agent. In fact, the residual hydroxyl surface concentration after silanization is even lower than $C_{limit}$, which accelerates the reaction-the achieved polymerization kinetic is similar to freshly 10 h calcinated particles. Therefore, the simultaneous rehydroxylation and silane grafting reactions offer an adequate silanization process that prevents excessive rehydroxylation.

The characteristics of the PA6 crystalline phase in the composites are summarized in Table 3. For both glass surface treatments, the crystalline structure formed during the polymerization of PA6 exhibits a higher melting temperature and melting enthalpy (1st heating), compared to the results obtained for the melt crystallized samples (following the 2nd heating). This was already reported in [27] for the bulk polymerization of PA6. Here, the presence of glass fillers did not change this effect, and a sharp, intense melting peak is obtained (see Appendix A for the DSC crystallization and melting curves). The matrix crystallization process during PA6 synthesis in the composite containing the silanized particles is faster compared to the composite prepared with calcinated particles. In addition, both the melting and crystallization temperatures, and melting enthalpies, increase with silane grafting. This could be due to the particles/matrix interactions at the interface in the case of silanized particles, modifying surface-induced nucleation. With a higher amount of surface hydroxyls, calcinated particles could also have modified the activator efficiency, and it is, therefore, possible that different particle surface chemistries resulted in different PA6 molecular weights and distributions. Further investigation will be required to identify the role of silane treatment on the polymerization and crystallization processes.

#### 3.3.2. Relevance of the Developed Protocol

In order to confirm the relevance of the established protocol, the silanization time was increased to 6 h. Then, the 6 h silanized particles were characterized by FTIR spectroscopy and compared to the 2 h silanized particles in Figure 8a.

It shows a significant increase in the intensity of the hydroxyl peak with increasing treatment time. This means that the grafting of the silane on the particle’s surface reaches a maximum and then stabilizes. Increasing the treatment time leads to a regeneration of hydroxyl groups inaccessible to the silane chains, which remain on the surface and can negatively impact polymerization. This is confirmed by the DSC thermograms illustrated in Figure 8b. Composites were synthesized with calcinated + 6 h silanized particles, and the polymerization/crystallization kinetics were compared to composites containing calcinated + 2 h silanized particles. The trapped and unreacted -OH groups indeed slowed down the polymerization and crystallization processes.

Overall, particles calcinated for 10 h, followed by a 2 h silanization treatment, represent the optimized conditions for the synthesis of reactive PA6/glass particles thermoplastic composites by anionic ring-opening polymerization. This protocol allows for significant silane grafting on the glass surface while limiting the presence of unreacted hydroxyl groups, which slow down or even deactivate the reaction. This ultimately allows the polymerization of PA6 with reaction features quite comparable to the neat resin.

### 3.4. Mechanical Properties

The mechanical properties of PA6/Glass composites presented in Table 4 show that the silane grafting protocol improves the performance of the composites in addition to controlling the polymerization reaction.

The use of calcinated particles results in lower strength and strain at break compared to silanized particles, although it allows achieving PA6 polymerization and provides a slightly higher modulus. In this case, the lower strength and strain at break indicate a weak interfacial adhesion. Indeed, the calcinated particle surface does not contain any coupling agent. Therefore, the interfacial adhesion is only resulting from van der Waals type interactions between the particles surface and the PA6 matrix. In the case of aminosilane treated particles, the entanglement of the grafted chains with the PA6 chains along with the formation of hydrogen bonds leads to the creation of strong physical interphase. This results in higher strength and strain compared to the previous case.

The heterogeneous composition of the interphase is however likely to be the cause of the lower modulus of composites containing silanized particles. The entanglement in the interphase can locally disrupt the crystallization ability of the matrix and hence decrease the modulus of the resulting composite.

A full study about the effect of glass surface chemistry and the associated interfacial interaction on the mechanical properties of the resulting composites will be the subject of a forthcoming work.

## 4. Discussion

In order to quantify the effect of the calcination time on the evolution of the hydroxyl groups’ surface concentration and density, the calcination temperature was fixed at 450 °C. Young [17] has studied the influence of the calcination temperature on the number of condensed hydroxyl groups (during dehydroxylation) that could be regenerated (rehydroxylation). He observed that rehydroxylation was only possible up to some temperature threshold during calcination, above which the condensation of some hydroxyl groups became irreversible and the number of hydroxyl groups that could be regenerated decreased. More specifically, the maximum number of dehydroxylated sites that could be reversibly rehydroxylated was reached at 450 °C. This tendency was confirmed by Hair [41], who studied the evolution of the hydroxyl groups’ surface density as a function of the calcination temperature by IR spectroscopy. Zhuravlev [15,18] has also identified the temperatures of dehydration, dehydroxylation and rehydroxylation, and demonstrated that dehydroxylation and siloxane formation could occur over a temperature range comprised between 200 °C and 400–500 °C, while full regeneration of hydroxyl groups in this range could be completed. Above this temperature range, the concentration of hydroxyl groups decreased and only partial regeneration was possible. Therefore, calcination at 450 °C does not allow to reach complete dehydroxylation, but it allows maximum reversible dehydroxylation. Such a temperature choice thus preserves all of the hydroxyls that can be rehydroxylated, which is an important consideration for the silanization process. Indeed, for the synthesis of anionic PA6-based composites, what is required is a calcination temperature and time resulting in an initial level of dehydroxylation that minimally affects the polymerization process. Accordingly, it is not critical to remove all hydroxyls from the surface, but instead to reach a state that offers a balance between the dehydroxylation and rehydroxylation processes, for the subsequent silane grafting step.

For the fully rehydroxylated particles (Figure 2), stabilization of the mass loss after about 1.5 h of dehydration shows that all physically adsorbed water has been removed. Furthermore, the mass stabilization shows that dehydroxylation has not yet started at 150 °C. Therefore, this drying condition is adequate as it leads to complete dehydration while avoiding the onset of dehydroxylation (around 200 °C) [18]. This dehydration temperature thus ensures a clear separation between the processes of evaporation of physically adsorbed water, and silanol condensation, which provides accuracy and reliability for the calculation of the hydroxyl groups’ surface concentration value.

The creation of hydrogen bonds between adjacent hydroxyls can proceed via different paths, leading to two categories of hydroxyls: (1) Hydroxyls linked via one hydrogen bond or more. These hydroxyls show a very large absorption peak, with a maximum below 3600 cm^−1^ In this case, the wave numbers of the hydroxyls both giving and accepting protons are lower compared to hydroxyls only giving a proton, due to a cooperative effect; (2) Hydroxyls able to create an additional hydrogen bond, with a wavenumber at 3720 cm^−1^, for terminal hydroxyls, and 3742 cm^−1^ for free geminal hydroxyls without any hydrogen bonding interaction.

From the IR spectroscopy results, all particles, regardless of the calcination time or rehydroxylation conditions, seem to have hydrogen-bonded hydroxyls (Figure 4a). This is consistent with the literature results since it has been shown that dehydroxylation continues up to 1000 °C [17,18]. Therefore, -OH groups may remain on the surface after calcination at 450 °C and may bond to each other if they are close enough. Our results show that the -OH groups remaining on the surface interact via hydrogen bonding since the wavenumber of the particles, regardless of the treatment, is below 3600 cm^−1^. Thus, the dehydroxylation process decreases the number of -OH groups, resulting in a gradual decrease in peak intensity and number of hydrogen bonds, before the wavenumber finally shifts to higher values.

When analyzing the thermograms of silanized particles, the total mass loss is not only due to the decomposition of surface grafted silanes, but also to dehydroxylation. It is possible to deduce the number of hydroxyls on the surface of each sample from the IR spectra results, by combining Equations (Equation 3) and (Equation 4) in Equation (Equation 5). We can then calculate the mass fraction associated with the loss of hydroxyls during the thermal measurements, in order to separate it from the mass loss related to the decomposition of the surface bonded silanes. This allows us to determine the real grafting degree and to tune the surface modification process.

The DSC thermograms show that the polymerization and crystallization onsets for the different composites are always delayed compared to the neat resin. This could be due to the presence of the glass particles in the reactive mixture limiting the mobility of the reactive species. The improvement of both polymerization and crystallization kinetics with the silanized particles, however, supports the compatibility of the silane-modified particles with the resin. Indeed, the presence of primary amine groups in the AEAPTMS silane could lead to the formation of hydrogen bonds at the particle-matrix interface, between the carbonyl of PA6 and the primary amines of the silane [36], resulting in enhanced strength and elongation at break.

Accordingly, the grafted amino-silane on the glass particles not only eliminates surface bonded hydroxyls that slow down the polymerization process, but also improves particulate-matrix adhesion by the creation of interfacial bonds [42]. It is thus interesting to note that the silanized calcinated particles, and the strictly calcinated particles, display similar polymerization rates (Figure 6b) since the grafted amino silane does not participate in the polymerization reaction itself. However, the crystallization is slightly faster with silanized particles, which could be due to the enhanced interactions with the matrix (Figure 6b, Table 3). In addition, it has been demonstrated that grafted silanes can contribute to crystal nucleation [43]. Therefore, this contribution of the silane, in promoting PA6 crystallization, increases the melting temperature and enthalpy.

The effect of various silane types on the resulting polymerization, crystallization and mechanical properties of the composites will be addressed in a forthcoming work.

## 5. Conclusions

Silicate-based materials as reinforcements for PA6 are ubiquitous in various application fields and yet, to date, fast anionic PA6 polymerization in such composites remains extremely difficult due to the disruptive presence of hydroxyl groups at the silicate surface. This work demonstrates that carefully tailoring and monitoring the mass concentration and density of surface hydroxyls of reinforcing particulates is necessary in order to optimize the processing of reactive anionic polyamide 6-based composites-an aspect that had been barely explored until now. This contribution will thus have an important impact on the preparation of anionic PA6-based composites not only with glass particles, which are used here as a model system, but also for glass fibers and other particles bearing hydroxyl groups at their surface. In that case, it is critical to determine the initial surface concentration in hydroxyl groups, and to precisely monitor both the rehydroxylation and silane grafting reactions during the silanization process. This fine control of -OH groups allows to maximize the PA6 polymerization reaction rate and crystallization-failing to do so can ultimately result in the complete inhibition of the polymerization reaction due to catalyst deactivation.

The hydroxyl groups’ surface concentration is systematically characterized and quantified by TGA and FTIR, for the complete particle surface modification sequence, from the dehydration, dehydroxylation and rehydroxylation processes, to the silanization step, as a function of treatment time and conditions-allowing to establish a direct relationship between FTIR transmittance and hydroxyls surface concentration. The effects of hydroxyl surface concentration after both dehydroxylation and rehydroxylation, and residual hydroxyl surface concentration after silanization, show that beyond 0.2 mmol OH·g^−1^ the polymerization reaction of PA6 is slowed down. This reaction can be completely inhibited when the hydroxyl concentration reaches 0.77 mmol OH·g^−1^ as in the case of fully rehydroxylated particles or raw pristine particles without further treatment. After a rigorous quantification and monitoring of the rehydroxylation process, we have demonstrated that, interestingly, both the rehydroxylation and silanization processes can be realized simultaneously without any negative impact on the polymerization reaction. This can be achieved with a silanization time of 2 h under the treatment conditions of the study. In this case, the silane agent gradually replaces the regenerated hydroxyls, removing one processing step. This work provides a roadmap for the preparation of reinforced reactive thermoplastic materials with a general approach adaptable to a variety of systems.

If tailoring the hydroxyl concentration on glass surfaces ensured the proper polymerization of ϵ-caprolactam, it may have an effect on the moisture uptake and the fiber-matrix adhesion in PA6 composites. Further investigation will be required for assessing the contribution of hydroxyl concentration and density, silane type and the associated interfacial interactions on the ageing behavior and resulting mechanical performances. This study should also consider the influence of hydroxyl groups and silane treatment on the crystallization behavior, as interfacial interactions can influence the nucleation process.

## Figures and Tables

**Figure 1 polymers-14-03663-f001:**
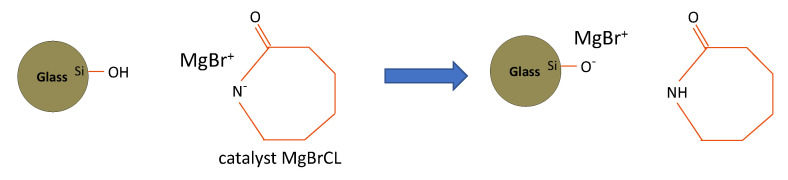
Deactivation, by surface hydroxyls, of the catalyst used for the synthesis of PA6 by anionic ring-opening polymerization.

**Figure 2 polymers-14-03663-f002:**
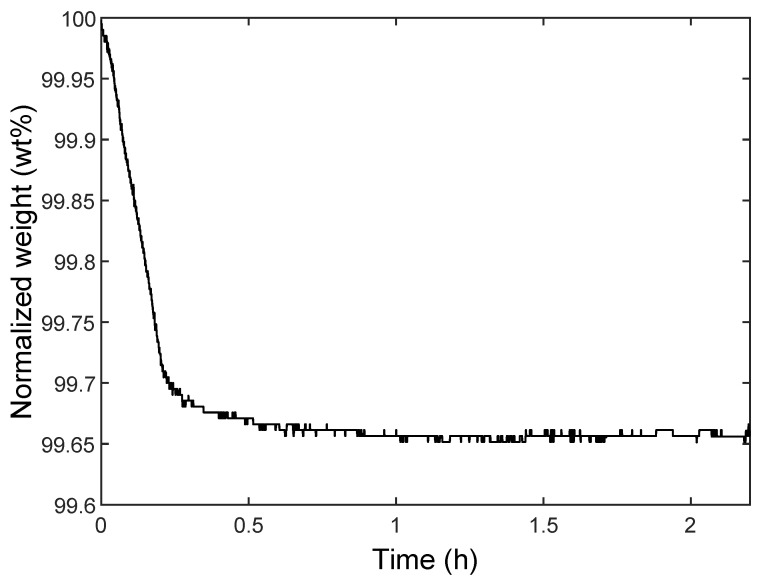
Thermogram of the normalized weight of the fully rehydroxylated particles in acid, heated at 150 °C for 2 h. The mass loss corresponds only to the physically adsorbed water at the surface of the particles since the onset of dehydroxylation has not been reached yet.

**Figure 3 polymers-14-03663-f003:**
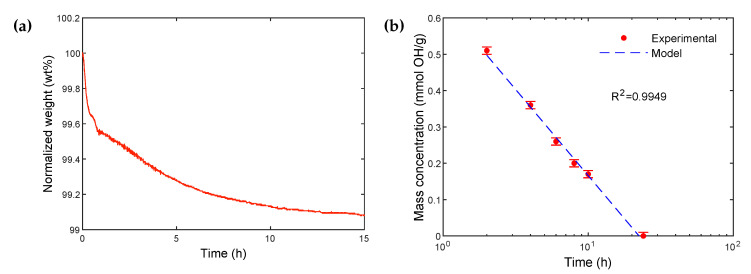
(**a**) TGA thermogram showing the normalized weight of fully rehydroxylated particles heated and maintained at 450 °C for 24 h. The mass loss corresponds to the total value of the desorption of physically adsorbed water and dehydroxylation processes. The mass loss related to physically adsorbed water is subtracted from the total mass loss to determine the amount related to the dehydroxylation process, in order to obtain the hydroxyl groups’ surface density. (**b**) Evolution of the hydroxyl group mass concentration as a function of calcination time, for the fully rehydroxylated particles (logarithmic scale). The dotted blue line is a linear regression.

**Figure 4 polymers-14-03663-f004:**
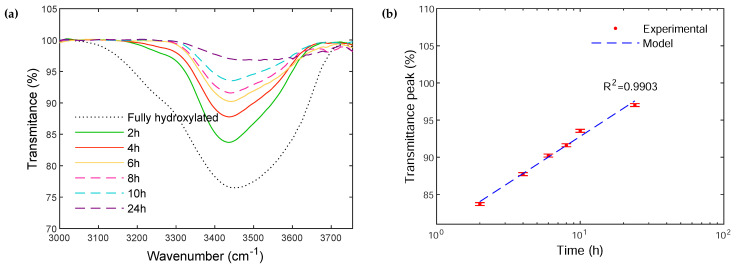
(**a**) Infrared spectra of initially rehydroxylated glass particles, as a function of the subsequent calcination time at 450 °C. (**b**) Evolution of the transmittance peak *T* as a function of calcination time at 450 °C. The dotted blue line corresponds to the linear regression of the Transmittance as a function of log(t) for the peak located in-between 3430 cm^−1^ and 3500 cm^−1^.

**Figure 5 polymers-14-03663-f005:**
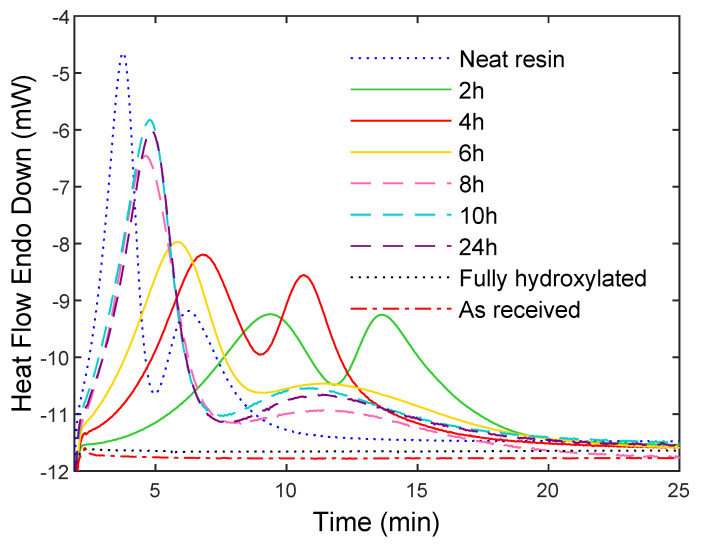
Comparison of polymerization and crystallization heat flow curves of the pure resin during isothermal synthesis at 180 °C with that of PA6/glass particle composites containing particles calcined for different times ranging from 0 h (fully rehydroxylated) to 24 h.

**Figure 6 polymers-14-03663-f006:**
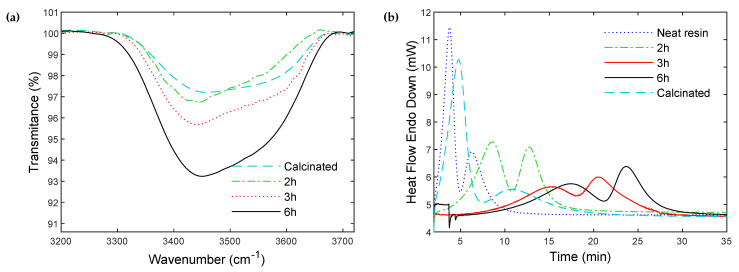
(**a**) FTIR spectra of glass particles calcinated for 10 h and subsequently partially rehydroxylated in the silanization solution (without the silane) for different durations. (**b**) Polymerization and crystallization heat flow curves during isothermal synthesis at 180 °C of PA6-glass particles composites, synthesized with partially rehydroxylated particles for 2 h, 3 h and 6 h in the silanization solution (without the silane agent), compared to the DSC curves of the pure resin and PA6-glass particles composites synthesized with the calcinated particles.

**Figure 7 polymers-14-03663-f007:**
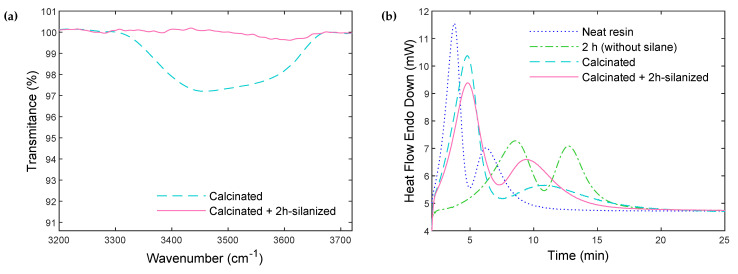
(**a**) FTIR spectra of calcinated particles, and of calcinated + 2 h-silanized particles. (**b**) Polymerization and crystallization heat flow curves during isothermal synthesis at 180 °C of PA6 and PA6-glass particles composites synthesized with calcinated particles, rehydroxylated particles for 2 h in the silanization solution (without silane) and calcinated + 2 h-silanized particles.

**Figure 8 polymers-14-03663-f008:**
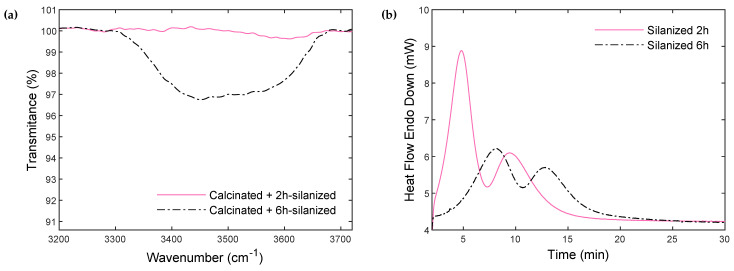
(**a**) FTIR spectra of 2 h and 6 h silanized particles. (**b**) Polymerization and crystallization heat flow curves during isothermal synthesis at 180 °C of composites synthesized with particles silanized for 2 h and 6 h.

**Table 1 polymers-14-03663-t001:** TGA weight loss and hydroxyls surface concentration, for particles calcinated for 10 h, and then partially rehydroxylated in the silanization solution (without the silane) for 2 h, 3 h and 6 h.

Treatment Time in Silanization Solution (without Silane)	TGA Weight Loss ^1^ (±0.01%)	Hydroxyls Mass Concentration *C_OH_* (±0.01 mmol OH·g^−1^)
0 h (calcinated)	0.14	0.16
2 h	0.19	0.21
3 h	0.24	0.27
6 h	0.31	0.34

^1^ Physically adsorbed water has been removed from the weight loss, leaving only the dehydroxylation process.

**Table 2 polymers-14-03663-t002:** TGA mass loss relative to the hydroxyls and silanes, for particles initially calcinated then silanized for 2 h, compared to strictly calcinated particles, and associated grafted silane mass concentration.

Particle Type	Total Weight Loss (± 0.01%)	Weight Loss Related to Hydroxyls (±0.01%)	Weight Loss Related to Silane (±0.01%)	Silane Mass Concentration $C_{\mathrm{silane}}$ (±mmol·g^−1^)
Calcinated	0.14	0.14	None	None
Calcinated + silanized for 2 h	0.71	0.05	0.66	0.05

**Table 3 polymers-14-03663-t003:** Thermal properties of the crystalline phase of PA6-glass particles composites synthesized with calcinated particles and calcinated + 2 h-silanized particles: Time at which the crystallization peak is observed during the isothermal step of polymerization and crystallization at 180 °C, and subsequent PA6 melting/crystallization temperatures, and melting enthalpies, associated to the heat-cool-heat experiments after composite synthesis.

	Isothermal Synthesis	1st Heating		Cooling	2nd Heating	
Particle Type	Peak of Crystallization (±0.4 min)	Melting Temperature	Melting Enthalpy Δ*H_m_*	Crystallization Temperature	Melting Temperature	Melting Nnthalpy Δ*H_m_*
		*T_m_* (±0.2 °C)	(±0.6 J.g^−1^)	*T_c_* (±0.3 °C)	*T_m_* (±0.2 °C)	(±0.6 J.g^−1^)
Calcinated	9.16	213.1	67.4	156.1	201.7	53.0
Calcinated + silanizedfor 2 h	7.83	215.6	71.2	159.4	206.7	54.7

**Table 4 polymers-14-03663-t004:** Average strength and strain at break of PA6/glass particles composites specimens containing calcinated particles and calcinated + 2 h-silanized particles.

Particle Type	Maximum Stress (MPa)	Strain at Break (%)	Tensile Modulus (MPa)
Calcinated	46 (±4.91)	1.8 (±0.22)	4390 (±84)
Calcinated + silanized for 2 h	63 (±4.09)	3.8 (±0.31)	3281 (±230)

## Data Availability

The data supporting reported results are available on request from the corresponding author.

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
