# Peer review of "Tailoring the Hydroxyl Density of Glass Surface for Anionic Ring-Opening Polymerization of Polyamide 6 to Manufacture Thermoplastic Composites"

_polymers, 2022, doi:10.3390/polym14173663_

Round 1

Reviewer 1 Report

The authors proposed the technique to controlled the functionalized of glass as a filler in PA6 composites which were prepared in situ. Although the analysis of chemical modification part is thoroughly done and the polymers were successfully polymerized, why the prepared specimens as well as their properties were not shown. How to justify that the current technique can improved the properties of the prepared specimens as composites? For example, the mechanical properties of the composites should be investigated to reflect that the employed technique has impacts on end uses.

Author Response

Dear Editor,

Please find attached our detailed response to the reviewers’ comments and a revised version of our manuscript entitled “Tailoring the hydroxyl density of glass surface for anionic ring opening polymerization of Polyamide 6 to manufacture thermoplastic composites”, submitted for publication in Polymers, Manuscript ID: polymers-1866134. You will find that we have brought modifications in answer to the reviewers’ comments. The additions to the text and modifications are highlighted in yellow and the excluded text is highlighted in red. Changes are identified in both the manuscript (with apparent modifications) and in this letter.

We thank the Reviewers for their constructive comments, and we respond point by point to their comments below.

Reviewer #1 general comments

The authors proposed the technique to controlled the functionalized of glass as a filler in PA6 composites which were prepared in situ. Although the analysis of chemical modification part is thoroughly done and the polymers were successfully polymerized, why the prepared specimens as well as their properties were not shown. How to justify that the current technique can improved the properties of the prepared specimens as composites? For example, the mechanical properties of the composites should be investigated to reflect that the employed technique has impacts on end uses.

Response

We would like first to thank the Reviewer for his/her appreciation of our work.

Comment #1 from Reviewer #1

  1. Why the prepared specimens as well as their properties were not shown.

Response

Our manuscript focuses on developing a methodology to control the surface density of hydroxyl groups at the surface of glass particle reinforcements, before and after silanization, in order to avoid the inhibition of both polymerization and crystallization processes of PA6-based composites by anionic ring-opening polymerization (AROP). We demonstrate first that there is a minimum surface density of hydroxyls over which polymerization inhibition becomes significant (Figure 5). It is then possible to control and monitor the dehydroxylation, rehydroxylation and silanization processes to ultimately control the hydroxyl surface density - both before silanization, and the residual hydroxyl density after silanization – to simultaneously prevent polymerization inhibition, and allow silane grafting at the particles surface (Figures 6 and 7).

The in situ synthesis of the composite specimens was performed in DSC in order to investigate the influence of hydroxyl groups on the polymerization and crystallization kinetics, by proposing a surface preparation protocol based on a rigorous analysis. After isothermal synthesis, several cooling-heating cycles were performed in order to analyze the crystallization behavior of the polymerized matrix as mentioned in the DSC experimental Section 2.4.2. We have reported the melting temperatures and enthalpies, and crystallization temperatures, of the composite specimens after polymerization, following heat-cool-heat DSC experiments in Table 3. We have also added the crystallization and melting curves from which the data in Table 3 were extracted, in the Supplementary data file (Figure S6, Section 5), and provided a discussion of these results in the last paragraph of Section 3.3 in order to better underline the possible relationship between the hydroxyls’ surface density and the crystalline behavior.

Therefore, we consider that it is not necessary to show larger specimens and their properties at this point.  We believe that presenting these aspects would significantly overextend the length of the manuscript.

Comment #2 from Reviewer #1

  1. How to justify that the current technique can improved the properties of the prepared specimens as composites? For example, the mechanical properties of the composites should be investigated to reflect that the employed technique has impacts on end uses.

Response

The main objective of this work was to control the hydroxyl groups surface density on the glass particles, during the silanization surface modification process, since hydroxyls can partially or completely inhibit the synthesis of PA6-based composites. We demonstrated that controlling the dehydroxylation, rehydroxylation and silanization processes allows polymerization and crystallization of the composites, with nearly identical features compared to the neat resin (Figure 7b). In contrast, pristine (as received) particles completely inhibit polymerization (Figure 5), illustrating the importance of controlling the hydroxyl groups surface density – this is the main contribution of this work.

Indeed, a partially inhibited polymerization significantly lowers the matrix properties by reducing the degree of conversion. Consequently, this decreases the final properties of the composite. Furthermore, the grafting of a silane coupling agent allows to improve the interfacial adhesion when the silane interacts with both the matrix and the reinforcing particles, as noted in the Introduction. In this work, the grafted aminosilane on the surface of the particles creates hydrogen bonds at the particle-matrix interface (as explained in the Discussion), which improves the interfacial adhesion and, therefore, enhances the properties of the resulting composite.

On the other hand, we demonstrated in Table 3 that the melting and crystallization temperatures, as well as the melting enthalpies of composites containing silanized particles are higher compared to composites prepared with calcinated particles. This could increase the mechanical properties of the composite. This aspect will be the subject of a forthcoming article investigating the effect of various types of silanes on the polymerization and crystallization processes and on the resulting mechanical properties of the composites. We believe again that adding a full mechanical properties analysis at this point would significantly extend the length of the manuscript.

We have modified the last sentence of the last paragraph of Section 3.3 in order to better underline that the role of the employed technique on the mechanical properties of the resulting composites requires further investigation.

“[…] Further investigation will be required to identify the role of silane treatment on the polymerization and crystallization processes and on the resulting mechanical properties of the composites. […]”

The final comment in the Conclusion Section also emphasizes the need for further study of the effect of glass surface chemistry and the associated interfacial interaction on the crystallization process and resulting mechanical properties.

Reviewer 2 Report

This work done by authors is interesting. This work provides a roadmap for the preparation of reinforced reactive thermoplastic materials with a general approach adaptable to a  variety of systems.

1.       Abstract reads like introduction. Authors are requested to make it should be crisp and clear.

2.       Authors are asked to give suitable applications where this research work could be translated. It could be included as a motivation in the Introduction Section.

3.       In the literature survey, clubbing of more number of references could be avoided. For instance, Ref. 22 – 33, 12-15, 31-35. Including pros and cons of each significant article may be helpful.

4.       Authors have done Particle surface modification. The impact of the particles surface chemistry on the polymerization and crystallization of the PA6/glass composites was quantified by DSC. What about its impact on mechanical and morphological properties of composites?

5.       This article provides a roadmap for the preparation of reinforced reactive thermoplastic materials.  What are the limitations of the present work, which could be taken further by research community?

6.       Conclusions are to be supported by the values of results.

Author Response

Dear Editor,

Please find attached our detailed response to the reviewers’ comments and a revised version of our manuscript entitled “Tailoring the hydroxyl density of glass surface for anionic ring opening polymerization of Polyamide 6 to manufacture thermoplastic composites”, submitted for publication in Polymers, Manuscript ID: polymers-1866134. You will find that we have brought modifications in answer to the reviewers’ comments. The additions to the text and modifications are highlighted in yellow and the excluded text is highlighted in red. Changes are identified in both the manuscript (with apparent modifications) and in this letter.

We thank the Reviewers for their constructive comments, and we respond point by point to their comments below.

Reviewer #2 general comments

This work done by authors is interesting. This work provides a roadmap for the preparation of reinforced reactive thermoplastic materials with a general approach adaptable to a variety of systems

Response

We would like to thank first the Reviewer for his/her appreciation of our work and the new understanding it brings. We respond point by point below to his/her questions.

Comment #1 from Reviewer #2

  1. Abstract reads like introduction. Authors are requested to make it should be crisp and clear.

Response

Again, we thank the Reviewer for his comments. In order to make the original contributions clearer and to highlight the innovation of our work, we have modified the abstract to make it crisper:

“[…]Thermoplastic composites are an option to face the challenge of the industry moving towards more sustainability. Reactive thermoplastics matrices are polymerized directly in contact with long fibers (glass, carbon, etc.) offer an ease of processing using well-known molding techniques (such as Resin Transfer Molding) due to their initially low viscosity with the same manufacturing processes as those conventionally used for thermosetting resins, such as Resin Transfer Molding. However, the surface chemistry of fibers directly impacts the polymerization. Indeed, it has been shown For Polyamide 6 (PA6)/glass composites, that the hydroxyl groups on the glass surface slow down the anionic ring opening polymerization (AROP) reaction, and can ultimately inhibit it, due to the labile protons that deactivate the catalyst. This work aims to thoroughly control the hydroxyl groups and the surface chemistry of glass particulates to facilitate in situ AROP - an aspect that has been barely explored until now. A model system composed of a PA6 matrix synthesized by AROP is polymerized reinforced with calcinated and silanized glass microparticles. The microparticles were calcinated, and then silanized in an acidic solution to graft 3-(2-Aminoethylamino)propyltrimethoxysilane (AEAPTMS) onto the remaining hydroxyl groups. We systematically quantify, by TGA and FTIR, the complete particle surface modification sequence, from the dehydration, dehydroxylation and rehydroxylation processes, to the silanization step. Finally, the impact of the particles surface chemistry on the polymerization and crystallization of the PA6/glass composites was quantified by DSC. The results confirm that a careful balance is required between the dehydroxylation process, the simultaneous rehydroxylation and silane grafting, and the residual hydroxyl groups, in order to maintain fast polymerization and crystallization kinetics, and to prevent reaction inhibition. Specifically, a hydroxyl concentration above 0.2 mmol OH.g-1 leads to a slowdown of the PA6 polymerization reaction. This reaction can be completely inhibited when the hydroxyl concentration reaches 0.77 mmol OH.g-1 as in the case of fully rehydroxylated particles or pristine raw particles. Furthermore, both the rehydroxylation and silanization processes can be realized simultaneously without any negative impact on the polymerization. This can be achieved with a silanization time of 2 h under the treatment conditions of the study. In this case, the silane agent gradually replacing the regenerated hydroxyls. This work provides a roadmap for the preparation of reinforced reactive thermoplastic materials. […]”

Comment #2 from Reviewer #2

  1. Authors are asked to give suitable applications where this research work could be translated. It could be included as a motivation in the Introduction Section.

Response

In this study, the control of the hydroxyl quantity during the calcination or the grafting of a silane coupling agent allows to improve composites manufacturing process by preserving the polymerization and crystallization kinetics during the "in-situ" synthesis of anionic PA6 based composite. We believe that the significance of this contribution is far-reaching. Glass based materials as reinforcements for PA6 are ubiquitous in various application fields such as structural parts for the automotive sector. Yet, to date, fast anionic PA6 polymerization in such composites is extremely difficult due to the disruptive presence of hydroxylated groups at the silicate surface. This contribution will have important impacts on the preparation of PA6-based composites not only with glass particles, which are used here as a model system, but also for glass fibers and other reinforcements bearing hydroxyl groups at their surface. We believe this to be an original and important contribution. Many other applications requiring accurate control of OH groups can benefit from this research work using the same protocol, such as composites manufacturing, glass coatings for hydrophobic surfaces, antibacterial and antifungal surfaces, waterproof windows, antifouling coatings for eyeglasses and any other application requiring silane treatment.

To make this aspect clearer, we added a comment in the Introduction Section:

[…] Finally, the impact of the silane grafting protocol is evaluated on the polymerization and crystallization kinetics of a model system composed of a PA6 matrix synthesized by anionic ring opening polymerization, reinforced by silanized glass microparticles, by means of DSC measurements. In our application, the control of the hydroxyl groups amount allows to preserve the polymerization and crystallization kinetics during the "in-situ" synthesis of anionic PA6 based composite. Many other applications can benefit from improved -OH groups control using the same protocol, such as composites manufacturing, glass coatings for hydrophobic surfaces, antibacterial and antifungal surfaces, waterproof windows, antifouling coatings for eyeglasses and any other application requiring silane treatment. […]

Comment #3 from Reviewer #2

  1. In the literature survey, clubbing of more number of references could be avoided. For instance, Ref. 22-33,12-15,31-35. Including pros and cons of each significant article may be helpful.

Response

We have avoided combining many references in the full paper as suggested by the Reviewer. This has been kept only if necessary.

Comment #4 from Reviewer #2

  1. Authors have done Particle surface modification. The impact of the particles surface chemistry on the polymerization and crystallization of the PA6/glass composites was quantified by DSC. What about its impact on mechanical and morphological properties of composites?

Response

As mentioned in the response to the Comment #1 and Comment #2 from Reviewer #1, the main objective of this work was to control the hydroxyl groups surface density on the glass particles, before and after silanization, in order to avoid the inhibition of both polymerization and crystallization processes of PA6-based composites by anionic ring-opening polymerization (AROP).

With the addition of glass particles, the crystallization process could be altered, especially due to a nucleating effect at the glass particles surface. The hydroxyls density at the surface of glass particles may also have an influence on the crystallization process. This is actually what the melting temperatures and enthalpies recorded upon heating by DSC (Table 3) indicate. In order to provide a better understanding about the crystal structure formed during the synthesis of PA6 in composites, we have reported the melting temperatures and enthalpies, and crystallization temperatures, of the PA6 based composites after polymerization, following heat-cool-heat DSC experiments in Table 3. We have also added the crystallization and melting curves from which the data in Table 3 were extracted, in the Supplementary data file (Figure S6, Section 5) and discussed these results in the last paragraph of Section 3.3 to underline the influence of the hydroxyl surface density on the crystal behavior.

The effect of various silane types and the resulting interfacial interactions on the polymerization and crystallization processes and on the mechanical properties of the composites will be the subject of a forthcoming article. We believe that adding a full mechanical properties analysis at this point would significantly extend the length of the manuscript.

Comment #5 from Reviewer #2

  1. This article provides a roadmap for the preparation of reinforced reactive thermoplastic materials. What are the limitations of the present work, which could be taken further by research community?

Response

The main limitation of the present study is the exact determination of the hydroxyl groups density. This calculation is tricky because of the uncertainty on the specific surface area measured by BET, as described in Section 3.1.1. Indeed, microporosities created by particle aggregates are not detectable by this technique, which considerably underestimates the value of the specific area and, therefore, results in excessive values of the hydroxyl groups density.

Comment #6 from Reviewer #2

  1. Conclusions are to be supported by the values of results

Response

We have modified the Conclusion Section and added some results values: 

[…] The hydroxyl groups surface concentration is systematically characterized and quantified by TGA and FTIR, for the complete particle surface modification sequence, from the dehydration, dehydroxylation and rehydroxylation processes, to the silanization step, as a function of treatment time and conditions - allowing to establish a direct relationship between FTIR transmittance and hydroxyls surface concentration. The effects of hydroxyl surface concentration after both dehydroxylation and rehydroxylation, and residual hydroxyl surface concentration after silanization, show that there is a critical value over which the PA6 polymerization reaction is slowed down and, ultimately, completely inhibited beyond 0.2 mmol OH.g-1 the polymerization reaction of PA6 is slowed down. This reaction can be completely inhibited when the hydroxyl concentration reaches 0.77mmol OH.g-1 as in the case of fully rehydroxylated particles or raw pristine particles without further treatment. After a rigorous quantification and monitoring of the rehydroxylation process, we have demonstrated that, interestingly, both the rehydroxylation and silanization processes can be realized simultaneously without any negative impact on the polymerization reaction. This can be achieved with a silanization time of 2 h under the treatment conditions of the study. In this case, the silane agent gradually replacing the regenerated hydroxyls, if the process is quantified and carefully monitored - removing one processing step. This work provides a roadmap for the preparation of reinforced reactive thermoplastic materials with a general approach adaptable to a variety of systems. […]

Round 2

Reviewer 1 Report

The authors should provide mechanical property characterization to make this manuscript complete.

Round 3

Reviewer 1 Report

The manuscript can now be accepted.